# Effects of Electrolyte Multivitamins and Neomycin on Immunity and Intestinal Barrier Function in Transported Lambs

**DOI:** 10.3390/ani14020177

**Published:** 2024-01-05

**Authors:** Cui Xia, Chunhui Duan, Conghui Chen, Xinyu Yang, Yingjie Zhang, Yueqin Liu, Yuzhong Ma

**Affiliations:** 1College of Animal Science and Technology, Agricultural University of Hebei, Baoding 071001, China; xiacui950114@163.com (C.X.); duanchh211@126.com (C.D.); zhangyingjie66@126.com (Y.Z.); 2College of Veterinary Medicine, Agricultural University of Hebei, Baoding 071001, China

**Keywords:** lambs, transport stress, electrolytic multivitamin, neomycin, immune function, intestinal health

## Abstract

**Simple Summary:**

Transport stress damages the body health and reduces the immunity of animals. Currently, drugs such as vitamins and antibiotics, etc., are used to alleviate transport stress. In this experiment, lambs were fed diets with electrolytic multivitamin and neomycin, respectively. The weight, stress hormones and immune indicators of sera were examined. At the same time, the villus height, crypt depth and the ratio of villus height to crypt depth (V/C) were measured. Furthermore, the mRNA expressions of Occludin and MUC1, and the protein expression of Occludin in the jejunal mucosa, as well as the mRNA expressions of ZO-1 and Occludin and the protein expression of Occludin in the colonic mucosa were examined. Otherwise, the mRNA expressions of TRAF6, TLR4, MyD88 and NF-kB, and the protein expressions of TLR4 and NF-kB p65, as well as the mRNA expressions of TRAF6, TLR4 and NF-kB in the colon were measured. Adding 375 mg/d/lamb electrolytic multivitamin in the basal diet from 2 d before transportation to 7 d after transportation could potentially improve the immunity and intestinal barrier function. This provides a reference for the application of anti-stress additives to alleviate transport stress of lambs.

**Abstract:**

Animals experience stress when they are transported. In this experiment, sixty 4-month-old lambs were randomly divided into three groups: CG (basal diet), EG (basal diet + 375 mg/d/lamb electrolytic multivitamin) and NG (basal diet + 200 mg/d/lamb neomycin). The transportation day was recorded as the 0th day. Blood, liver, spleen, jejunum and colon were collected on the 0th, 7th and 14th day. The results were as follows: In EG and NG groups, the lamb weights (*p* < 0.01), IgA and IgG (*p* < 0.05) increased significantly. The concentrations of ACTH, E, COR, IL-1β, IL-6 and IFN-γ decreased significantly (*p* < 0.01). The content of colonic propionate increased significantly (*p* < 0.05). The villus height and V/C increased, and crypt depth decreased significantly (*p* < 0.01). The mRNA expressions of Occludin and MUC1, and the protein expression of Occludin in the jejunal mucosa, the mRNA expressions of ZO-1 and Occludin, and the protein expression in the colonic mucosa increased significantly (*p* < 0.01). The mRNA expression of TRAF6 and the protein expression of TLR4 in the jejunum decreased significantly (*p* < 0.05), as well as the mRNA expressions of TLR4, MyD88 and NF-kB, and the protein expression of NF-kB p65 and the mRNA expressions of TRAF6, TLR4 and NF-kB in the colon (*p* < 0.01). In conclusion, an electrolytic multivitamin could potentially improve the immunity and intestinal barrier function, and when it was added with 375 mg/d in the basal diet for each lamb from 2 d before transportation to 7 d after transportation, it had a better effect than neomycin.

## 1. Introduction

Transport stress is an important factor in the production of stress in animals, and it inhibits the immune system, causes changes in body metabolism [1], and even causes serious damage to animal tissues and organs [2]. It also leads to weight loss and the suppression of the immune system in animals, as well as disrupting the stability of the gastrointestinal ecosystem of animals, causing changes in the bacterial community, releasing endotoxins, and weakening disease resistance, thereby increasing the risk of gastrointestinal infections [3]. It also affects animal welfare. Wei [4] found that transport stress could induce intestinal oxidative stress in pigs, destroy the antioxidant defense system, damage the intestinal barrier function, increase the risk of bacterial translocation, and lead to the incidence rate increasing. Therefore, the question of how to take effective measures to prevent and alleviate transport stress is crucial to improving animal immune function, reducing the incidence rate and improving animal welfare. So, the prevention and treatment of transport stress has become the focus of research. Currently, the drugs used to prevent and treat transport stress mainly include vitamins and antibiotics. Other researchers [5,6,7] found that adding a certain amount of vitamin C and electrolyte to the diet could alleviate anorexia, indigestion and diarrhea, and decrease fat and immunity and intestinal inflammatory reactions caused by stress in animals. It could effectively alleviate stress. In clinical practice, neomycin has good preventive and therapeutic effects on various animal diseases caused by bacteria. Transport stress often leads to a decrease in the immunity of animals, which can lead to a series of diseases after infection by pathogenic microorganisms. Therefore, the prevention and treatment of transport stress is extremely important. At present, the effect and mechanism of electrolytic multivitamin and neomycin in the immunity and intestinal barrier function of transported lambs are unclear.

Therefore, this experiment aimed to investigate the effects of different treatments on the stress hormone levels in sera, immune function and intestinal health of lambs by adding an electrolytic multivitamin before and after transportation, and adding neomycin after transportation. This experiment can provide a reference for the selection and application of anti-stress additives. It is meaningful for the rational utilization of animal resources and the improvement of animal welfare.

## 2. Materials and Methods

### 2.1. Experimental Design

Sixty male 4-month-old Hu lambs with similar weight were randomly divided into a control group (CG), electrolytic multivitamin (EG) and neomycin group (NG), respectively, with 20 for each group. The day of transportation was recorded as the 0th day. The lambs in the CG were fed with a basal diet, the lambs in EG had 375 mg/d electrolytic multivitamin (Anhui Lingmu Biotechnology Co., Ltd., Fuyang, China) added to the basal diet for each lamb from 2 d before transportation to 7 d after transportation, and the lambs in the NG were treated with neomycin (Henan Weilong Veterinary Medicine Co., Ltd., Kaifeng, China) 200 mg/d per lamb after transportation from 0 d to 7 d. The basic feed was provided by the sheep farm, and the lambs were fed twice a day at 7:00 and 17:00 to ensure that all lambs had leftover feed in their feed tanks every day. The composition and nutritional levels of the feed are shown in Table 1. On the 0th, 7th, and 14th days after transportation, five lambs were randomly selected from each group and weighed. Then, 5 mL jugular vein blood was collected for separating sera; it was placed in a sterile tube and frozen at −20 °C for further analysis.

Before feeding, first, an electrolytic multivitamin and neomycin were fully mixed with a small amount of feed, respectively. This ensured that the additives were fully consumed by each lamb. And then lambs could freely feed on basal diets. The lambs in each group were raised in their groups. The main components and contents of electrolytic multivitamins are shown in Table 2.

After blood collection, lambs were anesthetized with a dose of 0.2 mg/kg of xylazine (Hebei Mingeng Biotechnology Co., Ltd., Shijiazhuang, China) and then exsanguinated, and tissues were collected. The livers and spleens were collected and weighed for organ indices. The jejunum and colon tissues were taken out, and the contents were collected in sterile tubes and frozen in liquid nitrogen for volatile fatty acid (VFA) determination. And the middle part of the jejunum and colon was collected, and then the contents rinsed with saline and fixed in 4% formaldehyde solution for morphological examination. The remaining jejunum and colon samples were cut into two parts. One part of the jejunum and colon samples was stored in liquid nitrogen for later use. The other part of the jejunum and colon samples was used to collect the jejunal and colonic mucosa, which was scraped carefully with a glass slide and then stored in liquid nitrogen for later analysis.

### 2.2. Feeding and Management

This experiment was conducted from 1 June 2021 to 30 June 2021 at Zhangjiakou Lanhai Livestock Breeding Co., Ltd. (40°37′ N 115°03′ E and ~1300–1600 m above sea level, Zhangjiakou, China). The enclosure and breeding tools were thoroughly disinfected during the experiment period. Sixty male 4-month-old Hu sheep with similar weights were selected, and all experimental lambs were in three pens in the same building, with one pen for each group. The experimental lambs were transported for 8 h from 09:00 to 17:00 on June 18. They were transported locally in one truck, with a density of 1.7 m^2^/lamb. The speed of transportation was between 30 and 80 km/h. Driving sections included country roads and highways. On the day of transportation, the outdoor temperature was 18–29 °C and the relative humidity was 64%. There were no compartments inside the carriage, and the lambs were not tied down. During transportation, food and water was not provided and animals were allowed to drink within 12 h following transport.

### 2.3. Determination of Immune Function

#### 2.3.1. Determination of Immune Organ Index

The livers and spleens were weighed and the weights recorded. The organ index was calculated as follows:Immune organ index = Immune organ weight (g)/Live weight (kg)

Note: live weight refers to the weight of the lamb without feeding and drinking for 8 h.

#### 2.3.2. Assay of Sera Hormones and Immune Indices

Sera samples were thawed and mixed prior to testing. Concentrations of sera adrenocorticotropic hormone (ACTH), epinephrine (E), cortisol (COR), immunoglobulin A (IgA), immunoglobulin G (IgG), immunoglobulin M (IgM), and cytokines interleukin-1β (IL-1β), interleukin-2 (IL-2), interleukin-4 (IL-4), interleukin-6 (IL-6), interleukin-12 (IL-12), and interferon-γ (IFN-γ) were measured using enzyme-linked immunosorbent assay kits (Nanjing Jiancheng Bioengineering Institute, Nanjing, China) according to the standard procedures described by the manufacturer.

### 2.4. Determination of Intestinal Health

#### 2.4.1. Examination of Intestinal Morphology

Methods for determining intestinal morphology were performed according to Ma [8]. Remove the jejunal and colonic tissue samples fixed with 4% formaldehyde solution for more than 24 h, and then ethanol with increasing gradient concentrations is used for dehydration, xylene for cleared, and paraffin for permeation. The tissue samples were embedded with KD-BM III (Zhejiang Jinhua Kedi Instrument Equipment Co., Ltd., Jinhua, China). Several cross-sections were cut to a thickness of 5 µm and mounted on adherent slides, and an XDS-1B inverted biological microscope (Chongqing Guangdian Corporation, Chongqing, China) was used to observe the tissue morphology in the field of view at different magnifications. Three clear, intuitive visions of each tissue were selected for observation. Image J software v. 1 was used to measure the villus width, villus height, crypt depth, and the villus height-to-crypt depth ratio (V/C), and each field of view measured 2 data points per indicator.

#### 2.4.2. Examination of VFA

Assays for determining VFA were performed according to Zhang [9]. Samples of jejunal and colonic contents were thawed and thoroughly mixed immediately before testing. Concentrations of acetate, propionate, butyrate, and isovalerate were determined with the Daojin Gas Chromatograph System GC2010, as described in the instructions. Weigh 0.5 g of the jejunal and colonic contents into a centrifuge tube. Add 1.5 mL of ultrapure water, vortex-oscillate for 3–5 min, and centrifuge at 5000× *g* for 10 min. Take 1 mL of supernatant and transfer it to a centrifuge tube, then add 0.2 mL 25% metaphosphate solution, cover tightly, and shake well. And place it in an ice water bath for 30 min, and centrifuge at 10,000× *g* for 10 min. Take the supernatant and use a gas chromatograph to determine the concentration of VFA in the jejunal and colonic contents. The injection volume was 2 μL. The chromatographic column was an HP-INNOWAX (19091N-133) capillary column (30 m × 0.25 mm × 0.25 µm). The detector temperature was 220 °C. The column temperature was programmed to rise for 1 min at 80 °C, followed by a rise of 15 °C/min to 170 °C and was then maintained for 1.5 min. The carrier gas was high-purity nitrogen.

#### 2.4.3. Quantitative Real-Time PCR for Gene Expression Analysis

Total RNA was isolated from frozen jejunal and colonic mucosa and tissue (50 mg) with Trizol reagent (TransGen Biotech, Beijing, China). The concentration of total ribonucleic acid (RNA) in the samples was evaluated using a spectrophotometer (NanoDrop-2000, Thermo Fisher Scientific, Waltham, MA, USA) at 260 nm and 280 nm, respectively. Ratios of absorption (260:280 nm) between 1.8 and 2.0 for all samples were accepted as “pure” for RNA. RNA (1 µg) was used to generate copy DNA (cDNA) using the Reverse Transcription Kit according to the manufacturer’s instructions (Transgen Biotech, Beijing, China). Primer sequences for ZO-1, Occludin, MUC 1, MUC 2, TLR4, TRAF6, NF-kB, and MyD88 in jejunal and colonic mucosa and tissue were designed using the GenBank database from the National Center for Biotechnology Information and primer 5 design software. Quantitative real-time PCR was performed with SYBR Green Premix Es Taq (Takara, Beijing, China) using a StepOnePlus real-time PCR system (Applied Biosystems (Thermo-Fisher Scientific, Shanghai, China)) on 96-well plates with 20 µL of total reaction volume of 10 µL SYBR Green Premix, 1 µL cDNA, 0.4 µL of forward and 0.4 µL of reverse primers, and 8.2 µL of double-distilled water. Each reaction was run in duplicate. The PCR cycling protocol included one cycle of pre-incubation at 95 °C for 2 min, 40 cycles of denaturation at 95 °C for 5 s, and annealing at 60 °C for 30 s; each cycle was increased by 0.5 °C at 65–95 °C. β-actin was used as an internal control in this study. The average expression of the target genes relative to β-actin was determined using the 2^−ΔΔCt^ method, as described by Livak [10]. Primers for reverse transcription quantitative real-time PCR were synthesized by the Beijing Genomics Institution (BGI Genomics Co., Ltd., Beijing, China; Table 3).

#### 2.4.4. Western Blot Measured Protein Expression Levels

The protein was extracted from the jejunal and colonic mucosa and tissue using a protein extraction kit (Solarbio Biotech, Beijing, China) according to the manufacturer’s instructions. The protein concentration was determined as described above. Western blot analysis was carried out as previously described [11]. In brief, electrophoresis was performed with Sodium Dodecyl Sulfate-Polyacrylamide Gel Electrophoresis, followed by membrane transfer, blocking, the incubation of primary antibodies, membrane washing with tris buffered saline (TBS), the incubation of secondary antibodies, and film washing with TBS. Next, the membrane was automatically exposed and a picture was taken; its grayscale values were analyzed using Image J software v. 1 [12].

### 2.5. Statistical Analyses

All data were analyzed using single factor ANOVA in SPSS 21.0 statistical software [13] for the differences of indicators when using different treatments at the same time and the same treatment at different times. Duncan’s method for multiple comparisons was used. The General Linear Model (GLM) was used to analyze the effects of treatment and time. Effects *p* ≤ 0.05 were considered different. Values were expressed as “Mean ± standard error” (M ± SE).

## 3. Results

### 3.1. Weight and Immune Organ Index

From Table 4, the weight in EG was significantly higher than that in CG on day 7 (*p* < 0.01). The weights in NG and EG were significantly higher than that in CG on day 14 (*p* < 0.01). The weight of lambs gradually increased along with increasing time (*p* < 0.01).

### 3.2. Stress Hormones, Inflammatory Factors, and Immunoglobulins in the Sera

As shown in Table 5, on day 0, the concentrations of ACTH and COR in EG were significantly lower than those in CG and NG (*p* < 0.01). The concentration of E in CG was significantly higher than those in NG and EG (*p* < 0.01), and the concentration of E in NG was significantly higher than that in EG (*p* < 0.01). On day 7, the concentration of ACTH in EG was significantly lower than that in CG (*p* < 0.01). The concentration of COR in EG was significantly lower than those in CG and NG (*p* < 0.01). The concentrations of E in EG and NG were significantly lower than that in CG (*p* < 0.05). On day 14, the concentration of E in EG was significantly lower than those in CG and NG (*p* < 0.01). The concentration of COR in EG was significantly lower than that in CG (*p* < 0.01). The concentration of IgG in EG was significantly higher than those in EG and CG (*p* < 0.05). The concentrations of IgA in EG and NG were significantly higher than that in CG (*p* < 0.05). On day 0 and 7, the concentrations of IL-1β and IL-6 in EG were significantly lower than those in CG (*p* < 0.01). On day 7 and 14, the concentration of IFN-γ in EG was significantly lower than that in CG (*p* < 0.01). With the extension of feeding time, the concentrations of ACTH and E decreased significantly (*p* < 0.05), and the concentrations of COR, IL-1β, IL-2, IL-6, IL-12, and IFN-γ decreased significantly (*p* < 0.01). Otherwise, the concentrations of IL-4, IgA, IgG, and IgM increased significantly (*p* < 0.01).

### 3.3. VFA in the Jejunum and Colon

The contents of propionate in the colon of EG and NG were significantly higher than that in CG on day 7 (*p* < 0.05). The contents of acetate, propionate, butyrate, and isovalerate in the jejunum and colon of lambs gradually increased along with increasing time (*p* < 0.01).

### 3.4. Morphology of Intestinal Tissues

As shown in Table 6, Figure 1 and Figure 2, in the jejunum, on day 7, the crypt depth in EG was significantly lower than that in CG (*p* <0.01), and the V/C was significantly higher than that in CG (*p* < 0.01). On day 14, the villus height in EG was significantly higher than that in CG (*p* < 0.01). The crypt depths in EG and NG were significantly lower than that in CG (*p* < 0.01), and the V/C was significantly higher than that in the CG (*p* < 0.01). In the colon, on day 7, the villus height and V/C in EG and NG were significantly higher than those in CG (*p* < 0.01), and the crypt depth was significantly lower than that in CG (*p* < 0.01). The V/C in NG was significantly higher than that in CG on day 14 (*p* < 0.01). Otherwise, there was a tendency for interaction between treatment and time f for V/C in the colon. With the extension of time, the villus width in the jejunum increased significantly (*p* < 0.05). Otherwise, the villus width in the colon, villus height, and V/C in the jejunum and colon increased significantly (*p* < 0.01). The crypt depth in the jejunum and colon decreased significantly (*p* < 0.01).

### 3.5. Expression Levels of Important Genes in the Jejunal and Colonic Mucosa

As shown in Table 7 and Figure 3, in the jejunal mucosa, the mRNA expression of MUC1 on day 7 and the protein expression of Occludin on day 14 in EG was significantly higher than those in NG and CG (*p* < 0.01). The mRNA expression of Occludin on days 7 and 14 and the mRNA expression of MUC1 on day 14 in EG were significantly higher than that in CG (*p* < 0.01). In the colonic mucosa, the mRNA expressions of ZO-1 and Occludin in EG were significantly higher than those in CG on days 0 and 7 (*p* < 0.01). The mRNA expression of Occludin in EG was significantly higher than that in CG on day 14 (*p* < 0.01). The protein expression of Occludin in EG was significantly higher than those in CG and NG on days 0, 7, and 14 (*p* < 0.01). The mRNA expressions of ZO-1, Occludin, MUC1, and MUC2 and the protein expression of Occludin and MUC2 in the jejunal and colonic mucosa increased along with increasing time (*p* < 0.01).

### 3.6. Expression Levels of Important Genes and Proteins in the Jejunal and Colonic Tissues

From Table 8 and Figure 4, in the jejunum, the mRNA expressions of TLR4 and MyD88 in EG were significantly lower than those in CG on day 0 (*p* < 0.01). The mRNA expressions of TLR4, MyD88, and NF-kB and the protein expression of NF-kB p65 in EG were significantly lower than those in CG on day 7 (*p* < 0.01). On day 14, the mRNA expression of TRAF6 and the protein expression NF-kB p65 in NG and EG were significantly lower than those in CG (*p* < 0.05). The mRNA expressions of MyD88 and NF-kB in EG were significantly lower than those in CG (*p* < 0.01). In the colon, the mRNA expression of TRAF6 in EG was significantly lower than that in CG on day 7 (*p* < 0.01). The mRNA expressions of TLR4 and NF-kB in EG were significantly lower than those in CG on day 7 and 14 (*p* < 0.01). With the extension of time, the mRNA expressions of MyD88 in the jejunum, the mRNA expressions of TRAF6, TLR4, and NF-kB and the protein expressions of TLR4, NF-kB p65, and MyD88 in the jejunum and colon decreased significantly (*p* < 0.01).

## 4. Discussion

Animals are affected by various stressors during transportation: their catabolism increases, nutrient consumption accelerates, and immunity decreases. Gou et al. [14] found a positive correlation between transportation time and weight loss. Furthermore, Hongjian et al. [15] found that vitamin D could significantly improve animal production and alleviate stress symptoms. In this experiment, the weight of lambs in the EG and NG was significantly higher than that in the CG. On day 7, the weight in the EG increased compared to the CG. This indicated that electrolytic multivitamin and neomycin could effectively increase the weight of lambs and alleviate transport stress, and the electrolytic multivitamin had a better effect than neomycin. The possible reason was that the supplementation of vitamins could improve the immunity of lambs [16]. In addition, with the extension of time, the weight of lambs gradually increased within 0–14 days, and transport stress was gradually alleviated.

The relative weight of immune organs represents the development status of immune organs [17], which affects the immune function of the body. It was found that the liver weight of calves decreased significantly with 11 h transportation, which was due to an increase in vitamin consumption by stress [18]. The electrolytic multivitamin had a better effect on increasing immune organ indices than neomycin, and the possible reason was that the energy consumption increased after lambs were transported, which increased demand for vitamins in lambs; the electrolytic multivitamin could supplement vitamins to the body, so the immune organ indices increased [19,20].

Under stress, the function of the hypothalamic pituitary adrenal axis (HPA axis) in animals enhances, leading to an increase in the synthesis and secretion of related hormones such as ACTH, COR, and E [21]. Zeng et al. [22] found that COR levels in serum reduced significantly and the anti-stress of cows was improved by adding vitamins to the diet. In this experiment, the concentrations of ACTH, COR, and E in the EG were the lowest, and the concentration of E in the NG was significantly lower than that in the CG. This indicated that the electrolytic multivitamin and neomycin could alleviate transport stress effectively, and the effect of the electrolytic multivitamin was better than neomycin. With the extension of time, the concentrations of stress hormones such as ACTH, E, and COR in the sera of lambs decreased gradually, and transport stress alleviated gradually.

Inflammatory factors are important mediators in the neuroendocrine and immune networks. They participate in and regulate the physiological functions of the body, and play an important role in regulating immune and inflammatory responses [23]. It was found that various stress factors could induce the secretion of inflammatory factors such as IL-1β, IL-6, and TNF-α, which could lead to inflammatory reactions in the body [24]. Shojadoost et al. [19] found that adding an appropriate amount of vitamins to the diet could reduce the expressions of intestinal inflammatory factors, and it helped to keep a balance of intestinal inflammatory factors and improve the immune function of body. In this experiment, after adding the electrolytic multivitamin, the concentrations of IL-1β, IL-6, and IFN-γ in EG decreased significantly, which indicated that the electrolytic multivitamin could reduce the inflammatory reaction, which was helpful to alleviate transport stress. In addition, with the extension of time, the concentrations of IL-2, IL-6, IL-12, and IFN-γ in sera decreased gradually, and the concentration of IL-4 increased gradually. This indicated that there was still a serious inflammatory reaction in the lamb within 7 days after transportation, and the inflammatory reaction weakened and the stress alleviated gradually on day 14.

Immunoglobulins such as IgA, IgM, and IgG are the most common indicators for evaluating humoral immune function. Amaral et al. [25] found that stress could lead to the decrease of immunoglobulin in serum and the increase of the incidence rate in animals. Liu et al. [26] found that adding vitamin E to the diet could increase the content of immunoglobulin in the sera of chicken. In this experiment, the concentrations of IgA in the EG and NG were significantly higher than that in the CG. IgA plays an important role in intestinal mucosal immune response [27]. This indicated that the electrolytic multivitamin and neomycin could block the penetration of pathogenic microorganisms into the intestinal mucosa and exert immune regulatory effects. The concentration of IgA was the highest in EG, which indicated that the electrolytic multivitamin had a better effect than neomycin. It was also found that the concentration of IgG in the EG was significantly higher than those in NG and CG. This indicated that the electrolytic multivitamin could enhance the immunity of lambs effectively. The electrolytic multivitamin and neomycin had no effect on the concentration of IgM in sera; the specific reasons need further research. Stanger et al. [28] found that the immune cells returned to normal levels after the adult cattle landed for 6 days after they had been transported for 72 h. In this experiment, the concentrations of IgA, IgG, and IgM increased within 0–14 days, the transportation stress of lambs was gradually relieved, and the immune function of lambs enhanced gradually.

The VFA in intestinal contents is the degraded product of carbohydrates by intestinal microorganisms [29], which can regulate intestinal pH, inhibit the proliferation of harmful bacteria [30], protect the intestinal mucosal barrier [31], and provide energy for the body and intestinal cells [32]. In this experiment, the content of colonic propionate in the EG and NG increased significantly, and the possible reasons were related to the types of microorganisms in the intestine [33], the addition levels of the electrolytic multivitamin and neomycin, and the pH of the intestine. The addition of the electrolytic multivitamin and neomycin had no effect on the contents of acetate, butyrate, isovalerate in the jejunum and colon, as well as the propionate in the jejunum. It is possible that the majority of VFA was absorbed in the rumen, so the change in the intestine was not significant. With the extension of time, the contents of acetate, propionate, butyrate, and isovalerate in the jejunum and colon increased gradually. The digestive function of lambs enhanced gradually, and the stress response alleviated gradually.

The intestine is an important site for digestion, absorption, and nutrient metabolism. It can easily become a target organ for transport stress [34], which results in a series of digestive system diseases. Yu et al. [35] found that there was significant damage to the intestine during heat stress in pigs, including damage to the top of the intestinal villi, reductions in villi height, and a deepening of the crypt depth. Xiao et al. [36] found that vitamins could promote intestinal development, maintain the integrity of intestinal barriers, and improve the digestion and absorption of nutrients. In this experiment, the crypt depth of the jejunum in the EG decreased, and the V/C and villus height increased. Moreover, the crypt depth of the jejunum in the NG decreased, and the V/C increased. This indicated that the electrolytic multivitamin and neomycin could promote jejunal development, and the electrolytic multivitamin had more effects than neomycin. In the colon, the villus height and V/C of EG and NG increased, and the crypt depth in NG decreased. This indicated that the electrolytic multivitamin and neomycin could maintain the colonic morphological integrity of the lambs. The electrolytic multivitamin could improve the morphology in the jejunum and colon of lambs, and it improved the nutrient utilization efficiency and growth and development of lambs. This was consistent with the result showing that the electrolytic multivitamin could promote the weight of lambs.

Tight junction proteins play an important role in maintaining intestinal homeostasis [37], and the decrease of their expression levels results in an increase in diarrhea rate [38]. The mucous layer is the first line of defense against foreign pathogens, and the decrease in the content of mucin results in intestinal barrier dysfunction [39]. Carol et al. [40] found that vitamin D could increase the expressions of tight junction proteins such as Claudin, Occludin, and ZO-1 in intestinal mucosal epithelial cells, which protected the intestinal mucosal barrier. In this experiment, the mRNA expressions of Occludin and MUC1 of the jejunal mucosa in EG, as well as the protein expression of Occludin in EG, increased. This indicated that the electrolytic multivitamin could increase the mRNA expressions of Occludin and MUC1 and protein expressions of Occludin in the jejunal mucosa of lambs; it potentially enhanced tight junctions. In the colonic mucosa, the mRNA expressions of ZO-1 and Occludin in the EG, as well as the protein expression of Occludin in the EG, increased. This indicated that the electrolytic multivitamin had a potential effect on improving the barrier function of colonic mucosa. With the extension of time, the mRNA expressions of ZO-1, Occludin, MUC1, and MUC2, and the protein expressions of Occludin and MUC2 in jejunal and colonic mucosa, increased gradually, the tight junction recovered gradually, and the barrier function of the jejunal and colonic mucosa was enhanced. The electrolytic multivitamin had a better effect on protecting the intestinal digestive function, which is consistent with the change in the intestinal morphology of lambs. Therefore, it was speculated that the electrolytic multivitamin could maintain and improve physical and chemical barrier functions of the intestine by affecting the morphology of intestinal tissue, intestinal tight junction proteins, and mucins.

The TLRS/MyD88/NF-kB signaling pathway is an important pathway in the occurrence and development of inflammation, and it participates in biological processes such as immune regulation [41]. Yang et al. [24] found that the sustained activation of the TLR4/MyD88/NF-kB signaling pathway led to excessive immune inflammatory responses and ultimately damaged tissue. The expression levels of related signaling molecules in the TLR4/MyD88/NF-kB signaling pathway changed if an inflammatory response occurred. The electrolytic multivitamin downregulated the mRNA expressions of TRAF6, TLR4, MyD88, and NF-kB, and the protein expression of NF-kB p65 in the jejunum, and neomycin also downregulated the mRNA expression of TRAF6 and the protein expression of NF-kB p65. Meanwhile, the electrolytic multivitamin downregulated the mRNA expressions of TRAF6, TLR4, and NF-kB in the colon. This indicated that the electrolytic multivitamin could alleviate inflammation in the colon effectively. The mRNA expression levels of TRAF6, TLR4, MyD88, and NF-kB, and the protein expressions of TLR4, MyD88, and NF-kB p65 in the jejunum and colon, decreased along with increasing time, and the stress response of the jejunum and colon alleviated gradually. There was a consistent trend of change in the expression levels of TRAF6, TLR4, NF-kB, and MyD88 in the jejunum and colon, and the concentrations of IL-1β, IL-6, and IFN-γ in the sera of lambs. The electrolytic multivitamin was more beneficial than neomycin in reducing the inflammatory reaction. It is speculated that the electrolytic multivitamin may inhibit the expression levels of important genes in the TLR4/MyD88/NF-kB signaling pathway in the jejunum and colon of lambs, thereby inhibiting the production of inflammatory factors.

## 5. Conclusions

Dietary supplementation with a 375 mg/d/lamb electrolytic multivitamin had a negative effect on the immunity and intestinal barrier function of transported lambs. It has great potential to improve the effect of transport stress on immunity and intestinal barrier function.

## Figures and Tables

**Figure 1 animals-14-00177-f001:**
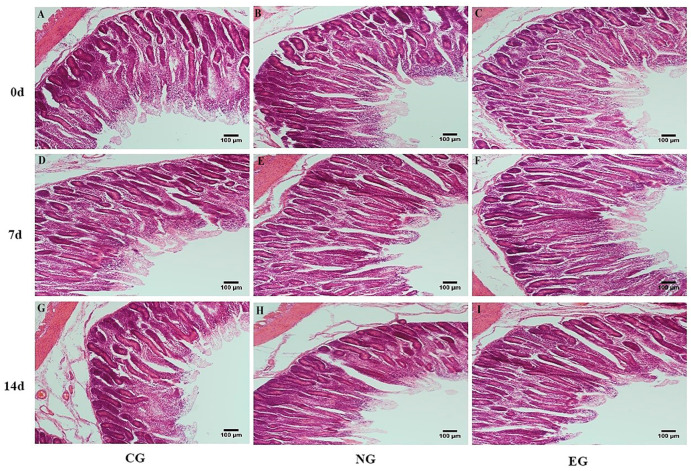
Effects of different treatments of transport stress on the jejunal morphology of lambs. (**A**) Jejunum of CG on day 0, (**B**) jejunum of NG on day 0, (**C**) jejunum of EG on day 0, (**D**) jejunum of control group on day 7, (**E**) jejunum of NG on day 7, (**F**) jejunum of EG on day 7, (**G**) jejunum of CG on day 14, (**H**) jejunum of NG on day 14, (**I**) Jejunum of EG on day 14.

**Figure 2 animals-14-00177-f002:**
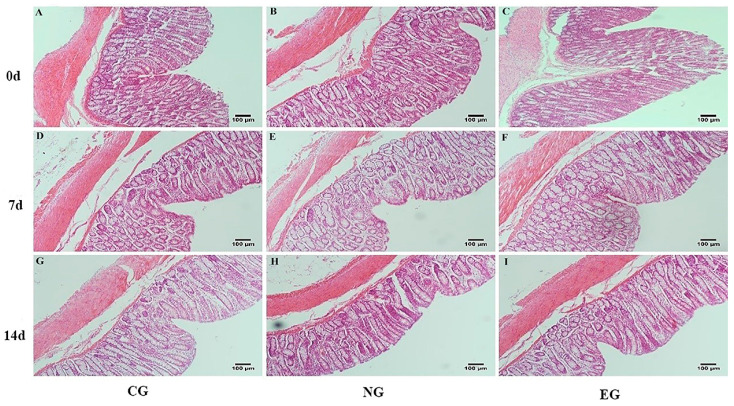
Effects of different treatments of transport stress on the colonic morphology of lambs. (**A**) Colon of CG on day 0, (**B**) colon of NG on day 0, (**C**) colon of EG on day 0, (**D**) colon of control group on day 7, (**E**) colon of NG on day 7, (**F**) colon of EG on day 7, (**G**) colon of CG on day 14, (**H**) colon of NG on day 14, (**I**) colon of EG on day 14.

**Figure 3 animals-14-00177-f003:**
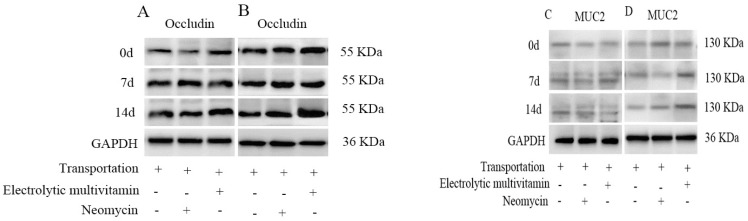
Effects of different treatments of transport stress on the protein expression of relevant genes in the jejunal and colonic mucosa. (**A**) The effect of different times and treatments on the protein expression of Occludin in the jejunal mucosa. (**B**) The effect of different times and treatments on the protein expression of Occludin in the colonic mucosa. (**C**) The effect of different times and treatments on the protein expression of MUC2 in the jejunal mucosa. (**D**) The effect of different times and treatments on the protein expression of MUC2 in the colonic mucosa. “+” represents lambs that were transported and treated with the addition of electrolytic multidimensional or neomycin, and “-” represents lambs that did not have the treatment addition. The original images are included in the Appendix A.

**Figure 4 animals-14-00177-f004:**
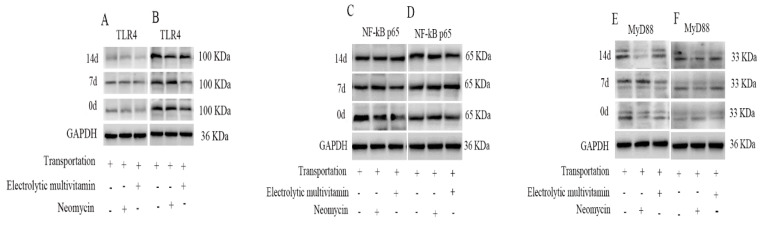
Effects of different treatments of transport stress on the expression levels of relevant genes in the jejunum and colon. (**A**) The effect of different times and treatments on the protein expression of TLR4 in the jejunum. (**B**) The effect of different times and treatments on the protein expression of TLR4 in the colon. (**C**) The effect of different times and treatments on the protein expression of NF-kB p65 in the jejunum. (**D**) The effect of different times and treatments on the protein expression of NF-kB p65 in the colon. (**E**) The effect of different times and treatments on the protein expression of MyD88 in the jejunum. (**F**) The effect of different times and treatments on the protein expression of MyD88 in the colon. “+” represents lambs that were transported and were treated with the addition of electrolytic multidimensional or neomycin, and “-” represents lambs that did not have the treatment addition. The original images are included in the Appendix A.

**Table 1 animals-14-00177-t001:** Composition and nutrient levels of the basal diet (dry matter basis).

Ingredients	Content %	Nutritional Level	Content
Cracked Corn	55.00	Neutral Detergent Fibers	33.33
Soybean meal	20.00	Crude Protein	18.06
Peanut seeding	12.50	Acidic Detergent Fibers	14.49
Peanut meal	9.00	Metabolic Energy (MJ/kg) ^2^	12.50
Premix ^1^	2.50	Ca	0.75
NaCl	0.50	NaCl	0.50
Baking soda	0.50	P	0.31
Total	100.00		

^1^ Each kg premix contained the following: VA 15356 IU, VD 4300 IU, VE 50 mg, Fe 88.70 mg, Zn 70.90 mg, Mn 51.80 mg, Cu 13.75 mg, Se 0.23 mg, I 1.50 mg, Co 0.49 mg. ^2^ Metabolic energy was a calculated value, while the others were measured values.

**Table 2 animals-14-00177-t002:** The main components and contents of electrolytic multivitamins.

Components	Contents
Vitamin C	≥2000 IU/kg
Vitamin B2	≥750 mg/kg
Vitamin A	30,000–5,000,000 IU/kg
Vitamin D3	75,000–20,000,000 IU/kg
Vitamin E	≥500 IU/kg
Vitamin B1	≥500 mg/kg
Water	≤10%
Folate	30 mg/kg
Taurine	20,000 mg/kg
Zn	1000 mg/kg
Mn	1000 mg/kg
Fe	1000 mg/kg
Cu	600 mg/kg

**Table 3 animals-14-00177-t003:** Gene primer information.

Primer	Sequence (5′→3′)		Length (bp)	Annealing Temperature (°C)	Accession Number
ZO-1	Forward	CATCACGCCAGCATACAA	177	52	XM_042235171
	Reverse	GCAGACTTCAGGAGGGTTT			
Occludin	Forward	CCAGCGTTGTAAGGTCAG	218	51	>XM_042233706.1
	Reverse	ATCCACGTAGAGTCCAGTAG			
MUC1	Forward	CTCCCTCACAGGACCCATCA	126	56	>XM_027976040.2
	Reverse	CGTTCCCACTCCCGTTTC			
MUC2	Forward	TCACCTGCCCTGACTTTGA	144	57	>DQ152978.1
	Reverse	TGTGCGAAATCTCCCTCGT			
TRAF6	Forward	TTCAACAGTTAGAGGGTCG	330	53	XM_015101111.3
	Reverse	GGGATATGTAGTTGGCACA			
TLR4	Forward	AAGGTGATTGTCGTGGTG	178	57	>NM_001135930.1
	Reverse	TGTTCAGAAGGCGATAGA			
MyD88	Forward	AGTACAAGCCAATGAAGAAAGA	100	58	>DQ152979.1
	Reverse	AGGCGAGTCCAGAACCAG			
NF-kB	Forward	CAACTGCTCTACCTCCTG	290	58	>XM_027966801.2
	Reverse	CTTCACTGTTACGGGTTC			
β-actin	Forward	CCCTGGAGAAGAGCTACGAG	98	58	NM-102179831
	Reverse	CAGGAAGGAAGGCTGGAAGA			

**Table 4 animals-14-00177-t004:** Effects of different treatments on weight and immune organ index of lambs.

Items	Time		Groups		*p*-Value		
		CG	NG	EG	Time	Treatment	Time × Treatment
Weight/kg	0 d	19.88 ± 0.16 ^B^	20.25 ± 0.35 ^C^	20.88 ± 0.38 ^B^			
	7 d	20.55 ± 0.40 ^Bb^	21.64 ± 0.40 ^Bab^	21.86 ± 0.25 ^Ba^	<0.010	<0.010	0.679
	14 d	22.13 ± 0.36 ^Ab^	23.48 ± 0.23 ^Aa^	23.62 ± 0.41 ^Aa^			
Liver index (g/kg)	0 d	17.47 ± 1.12	18.95 ± 1.39	17.31 ± 1.60			
	7 d	18.22 ± 1.84	19.07 ± 1.97	19.21 ± 1.52	0.562	0.814	0.977
	14 d	19.16 ± 2.14	19.37 ± 1.30	19.43 ± 1.44			
Spleen index (g/kg)	0 d	0.88 ± 0.06	0.94 ± 0.07	0.83 ± 0.09			
	7 d	0.89 ± 0.09	0.88 ± 0.10	0.89 ± 0.07	0.720	0.875	0.928
	14 d	0.87 ± 0.10	0.82 ± 0.05	0.83 ± 0.07			

Note: ^ab^ Different lowercase letters superscripted in the same row indicate a significant difference between different treatments (*p* < 0.05). ^ABC^ The superscript values in the same column with different uppercase letters indicate significant difference at different times (*p* < 0.05). The same letter or no letter indicates no significant difference (*p* > 0.05). There is no meaning between uppercase and lowercase letters.

**Table 5 animals-14-00177-t005:** Effects of different treatments of transport stress on stress hormones and immune indices in the sera of lambs.

Items		Time	Groups			*p*-Value		
			CG	NG	EG	Time	Treatment	Time × Treatment
Stress hormones	ACTH (ng/L)	0 d	56.91 ± 1.12 ^Aa^	53.91 ± 3.51 ^a^	46.33 ± 2.59 ^b^			
		7 d	53.56 ± 1.38 ^ABa^	49.71 ± 2.75 ^ab^	45.04 ± 1.32 ^b^	0.027	<0.010	0.943
		14 d	50.53 ± 1.98 ^B^	47.41 ± 2.84	43.27 ± 2.46			
	E (ng/mL)	0 d	1.03 ± 0.02 ^Aa^	0.84 ± 0.02 ^b^	0.75 ± 0.03 ^c^			
		7 d	0.91 ± 0.03 ^Ba^	0.80 ± 0.04 ^b^	0.73 ± 0.03 ^b^	0.031	<0.010	0.278
		14 d	0.87 ± 0.05 ^Ba^	0.83 ± 0.02 ^a^	0.70 ± 0.05 ^b^			
	COR (ng/mL)	0 d	47.07 ± 1.95 ^a^	44.25 ± 1.39 ^Aa^	39.15 ± 1.75 ^b^			
		7 d	45.85 ± 1.27 ^a^	43.89 ± 1.29 ^ABa^	38.08 ± 0.90 ^b^	<0.010	<0.010	0.979
		14 d	42.75 ± 2.47 ^a^	39.47 ± 1.40 ^Bab^	35.40 ± 1.34 ^b^			
Inflammatory factors	IL-1β (ng/L)	0 d	28.41 ± 1.40 ^a^	25.84 ± 1.19 ^Aab^	23.87 ± 1.04 ^Ab^			
		7 d	24.04 ± 1.79 ^a^	23.01 ± 1.50 ^Bab^	20.81 ± 0.78 ^Bb^	<0.010	<0.010	0.925
		14 d	22.94 ± 1.89	21.60 ± 0.90 ^B^	19.82 ± 1.05 ^B^			
	IL-2 (ng/L)	0 d	36.42 ± 1.42 ^A^	34.47 ± 2.40	33.30 ± 2.39 ^A^			
		7 d	30.13 ± 1.64 ^B^	31.79 ± 1.36	29.42 ± 1.32 ^AB^	<0.010	0.361	0.935
		14 d	28.60 ± 2.66 ^B^	27.43 ± 0.98	27.16 ± 1.35 ^B^			
	IL-4 (ng/L)	0 d	48.88 ± 1.63 ^B^	49.99 ± 4.34	48.27 ± 1.49			
		7 d	53.47 ± 2.17 ^AB^	55.54 ± 3.04	54.80 ± 3.26	<0.010	0.726	0.997
		14 d	57.28 ± 2.61 ^A^	57.64 ± 1.17	57.31 ± 3.14			
	IL-6 (ng/L)	0 d	60.26 ± 1.75 ^Aa^	56.99 ± 1.98 ^Aab^	51.51 ± 2.11 ^b^			
		7 d	56.03 ± 2.00 ^ABa^	53.93 ± 2.35 ^ABab^	47.31 ± 2.17 ^b^	<0.010	<0.010	0.876
		14 d	50.06 ± 2.60 ^B^	50.17 ± 1.83 ^B^	46.05 ± 1.15			
	IL-12 (ng/L)	0 d	160.67 ± 4.21	159.46 ± 3.86 ^A^	158.59 ± 3.27			
		7 d	154.22 ± 2.13	153.36 ± 1.77 ^AB^	151.15 ± 3.03	<0.010	0.416	0.958
		14 d	155.77 ± 3.77	148.78 ± 1.96 ^B^	148.47 ± 3.82			
	IFN-γ (ng/L)	0 d	69.12 ± 2.67	64.58 ± 2.97	61.32 ± 2.20 ^A^			
		7 d	65.14 ± 3.44 ^a^	60.32 ± 1.59 ^ab^	55.92 ± 2.92 ^ABb^	<0.010	<0.010	0.994
		14 d	60.53 ± 1.30 ^a^	57.69 ± 4.36 ^ab^	51.66 ± 1.82 ^Bb^			
Immunoglobulins	IgA (mg/mL)	0 d	1.09 ± 0.03	0.97 ± 0.06 ^B^	1.09 ± 0.09 ^B^			
		7 d	1.12 ± 0.16	1.33 ± 0.09 ^AB^	1.42 ± 0.03 ^A^	<0.010	0.033	0.105
		14 d	1.31 ± 0.05 ^b^	1.53 ± 0.06 ^Aa^	1.52 ± 0.09 ^Aa^			
	IgG (mg/mL)	0 d	11.29 ± 0.07	11.39 ± 0.62 ^B^	10.49 ± 0.43 ^B^			
		7 d	11.85 ± 0.71	12.30 ± 0.52 ^A^	13.14 ± 0.46 ^A^	<0.010	0.015	0.357
		14 d	12.19 ± 0.49 ^b^	12.63 ± 0.30 ^Ab^	13.80 ± 0.31 ^Aa^			
	IgM (mg/mL)	0 d	1.92 ± 0.09 ^B^	2.04 ± 0.10 ^B^	2.05 ± 0.09 ^B^			
		7 d	2.17 ± 0.10 ^AB^	2.22 ± 0.14 ^AB^	2.22 ± 0.13 ^B^	<0.010	0.138	0.602
		14 d	2.31 ± 0.06 ^A^	2.54 ± 0.07 ^A^	2.65 ± 0.06 ^A^			

Note: ^abc^ Different lowercase letters superscripted in the same row indicate significant differences between different treatments (*p* < 0.05). ^AB^ The superscript values in the same column with different uppercase letters indicate significant differences at different times (*p* < 0.05). The same letter or no letter indicates no significant difference (*p* > 0.05). There is no meaning between uppercase and lowercase letters.

**Table 6 animals-14-00177-t006:** Effects of different treatments of transport stress on the morphology of the jejunum and colon of lambs.

Items		Time	Groups			*p*-Value		
			CG	NG	EG	Time	Treatment	Time × Treatment
Jejunum	Villus width (µm)	0 d	131.29 ± 3.84 ^B^	133.26 ± 4.84	136.75 ± 6.90			
		7 d	136.65 ± 4.00 ^AB^	141.57 ± 4.74	140.37 ± 5.40	0.019	0.713	0.996
		14 d	147.02 ± 6.32 ^A^	148.36 ± 7.59	150.39 ± 10.36			
	Villus height (µm)	0 d	531.95 ± 11.19	546.68 ± 8.61	561.45 ± 10.46 ^B^			
		7 d	540.74 ± 14.59	565.68 ± 15.22	581.36 ± 13.40 ^AB^	<0.010	<0.010	0.796
		14 d	548.04 ± 13.27 ^b^	582.95 ± 12.16 ^ab^	609.21 ± 13.56 ^Aa^			
	Crypt depth (µm)	0 d	171.77 ± 8.88	166.81 ± 3.89 ^A^	154.29 ± 6.47 ^A^			
		7 d	165.48 ± 3.67 ^a^	154.05 ± 4.38 ^Bab^	144.77 ± 7.56 ^ABb^	<0.010	<0.010	0.840
		14 d	158.82 ± 2.62 ^a^	140.85 ± 4.09 ^Cb^	132.58 ± 5.79 ^Bb^			
	V/C ^1^	0 d	3.17 ± 0.22	3.29 ± 0.31 ^C^	3.69 ± 0.19 ^B^			
		7 d	3.27 ± 0.07 ^b^	3.69 ± 0.13 ^Bab^	4.11 ± 0.27 ^ABa^	<0.010	<0.010	0.547
		14 d	3.46 ± 0.10 ^b^	4.16 ± 0.13 ^Aa^	4.64 ± 0.20 ^Aa^			
Colon	Villus width (µm)	0 d	72.61 ± 3.63	77.40 ± 3.71	70.75 ± 3.00 ^B^			
		7 d	75.23 ± 2.82	86.60 ± 2.81	81.42 ± 2.38 ^A^	<0.010	0.051	0.298
		14 d	81.06 ± 2.32	82.79 ± 2.78	85.00 ± 2.41 ^A^			
	Villus height (µm)	0 d	188.86 ± 6.27	208.09 ± 11.95 ^B^	196.80 ± 7.60 ^B^			
		7 d	199.50 ± 4.35 ^b^	223.63 ± 4.91 ^Ba^	236.10 ± 10.90 ^Aa^	<0.010	<0.010	0.394
		14 d	222.31 ± 27.50	254.39 ± 20.09 ^A^	245.90 ± 34.12 ^A^			
	Crypt depth (µm)	0 d	51.59 ± 2.00	45.68 ± 1.16 ^A^	47.14 ± 1.55 ^A^			
		7 d	49.06 ± 2.74 ^a^	42.22 ± 1.21 ^Bb^	39.84 ± 1.13 ^Bb^	<0.010	<0.010	0.367
		14 d	46.49 ± 1.93	41.21 ± 0.93 ^B^	43.51 ± 1.17 ^AB^			
	V/C	0 d	3.72 ± 0.28	4.54 ± 0.22 ^C^	4.23 ± 0.27 ^B^			
		7 d	4.14 ± 0.24 ^b^	5.32 ± 0.16 ^Ba^	5.93 ± 0.57 ^Aa^	<0.010	<0.010	0.071
		14 d	4.87 ± 0.41 ^b^	6.28 ± 0.24 ^Aa^	5.62 ± 0.32 ^Aab^			

^1^ V/C = the ratio of villus height to crypt depth. Note: ^ab^ different lowercase letters superscripted in the same row indicate significant differences between different treatments (*p* < 0.05). ^ABC^ The superscript values in the same column with different uppercase letters indicate significant differences at different times (*p* < 0.05). The same letter or no letter indicates no significant difference (*p* > 0.05). There is no meaning between uppercase and lowercase letters.

**Table 7 animals-14-00177-t007:** Effects of different treatments of transport stress on the expression levels of important genes in the jejunal and colonic mucosa.

Items			Time	Groups			*p*-Value		
				CG	NG	EG	Time	Treatment	Time × Treatment
Jejunal mucosa	mRNA expression	ZO-1	0 d	1.14 ± 0.13 ^B^	1.30 ± 0.16 ^C^	1.46 ± 0.13 ^B^			
			7 d	1.83 ± 0.20 ^B^	1.95 ± 0.23 ^B^	2.01 ± 0.25 ^B^	<0.010	0.170	0.971
			14 d	2.41 ± 0.25 ^A^	2.60 ± 0.22 ^A^	2.88 ± 0.27 ^A^			
		Occludin	0 d	1.46 ± 0.12 ^B^	1.57 ± 0.19 ^B^	1.89 ± 0.20 ^B^			
			7 d	1.82 ± 0.16 ^ABb^	2.31 ± 0.18 ^Aab^	2.56 ± 0.20 ^Aa^	<0.010	<0.010	0.835
			14 d	2.20 ± 0.21 ^Ab^	2.61 ± 0.20 ^Aab^	2.95 ± 0.21 ^Aa^			
		MUC1	0 d	1.41 ± 0.15 ^B^	1.63 ± 0.15 ^B^	1.86 ± 0.20 ^B^			
			7 d	1.76 ± 0.11 ^ABb^	1.95 ± 0.14 ^ABb^	2.34 ± 0.16 ^Aa^	<0.010	<0.010	0.979
			14 d	2.01 ± 0.16 ^Ab^	2.21 ± 0.16 ^Aab^	2.60 ± 0.12 ^Aa^			
		MUC2	0 d	1.54 ± 0.14	1.63 ± 0.13	1.80 ± 0.15			
			7 d	1.70 ± 0.12	1.75 ± 0.13	1.83 ± 0.11	0.106	0.220	0.986
			14 d	1.79 ± 0.10	1.89 ± 0.10	1.94 ± 0.15			
	Protein expression	Occludin	0 d	0.64 ± 0.01 ^Bb^	0.84 ± 0.04 ^Ba^	0.91 ± 0.05 ^Ba^			
			7 d	0.86 ± 0.11 ^B^	0.93 ± 0.05 ^AB^	1.02 ± 0.06 ^B^	<0.010	<0.010	0.273
			14 d	1.09 ± 0.01 ^Ab^	1.04 ± 0.04 ^Ab^	1.30 ± 0.08 ^Aa^			
		MUC2	0 d	0.09 ± 0.004 ^B^	0.10 ± 0.02 ^B^	0.11 ± 0.02 ^B^			
			7 d	0.12 ± 0.01 ^AB^	0.12 ± 0.01 ^AB^	0.13 ± 0.02	<0.010	0.508	0.154
			14 d	0.12 ± 0.01 ^A^	0.13 ± 0.01 ^A^	0.13 ± 0.01			
		ZO-1	0 d	1.30 ± 0.14 ^Bb^	1.60 ± 0.12 ^Bab^	1.92 ± 0.21 ^Ba^			
			7 d	1.65 ± 0.16 ^Bb^	1.94 ± 0.18 ^Bab^	2.32 ± 0.18 ^ABa^	<0.010	<0.010	0.989
			14 d	2.25 ± 0.22 ^A^	2.53 ± 0.15 ^A^	2.75 ± 0.18 ^A^			
Colonic mucosa	mRNA expression	Occludin	0 d	1.79 ± 0.21 ^Bb^	2.02 ± 0.19 ^Bab^	2.42 ± 0.19 ^Ba^			
			7 d	2.06 ± 0.19 ^ABb^	2.43 ± 0.17 ^ABab^	2.83 ± 0.19 ^ABa^	<0.010	<0.010	0.979
			14 d	2.44 ± 0.17 ^Ab^	2.90 ± 0.17 ^Aab^	3.25 ± 0.16 ^Aa^			
		MUC1	0 d	1.49 ± 0.14	1.60 ± 0.12	1.65 ± 0.10 ^B^			
			7 d	1.60 ± 0.11	1.74 ± 0.10	1.77 ± 0.10 ^AB^	<0.010	0.136	0.999
			14 d	1.77 ± 0.11	1.89 ± 0.11	1.98 ± 0.10 ^A^			
		MUC2	0 d	1.51 ± 0.16	1.78 ± 0.19	1.89 ± 0.18			
			7 d	1.78 ± 0.19	1.80 ± 0.15	1.82 ± 0.23	0.679	0.497	0.915
			14 d	1.80 ± 0.18	1.86 ± 0.22	1.92 ± 0.14			
	Protein expression	Occludin	0 d	0.62 ± 0.05 ^Bb^	0.76 ± 0.05 ^Bb^	0.89 ± 0.04 ^Ca^			
			7 d	0.84 ± 0.07 ^Ab^	0.93 ± 0.05 ^Bb^	1.16 ± 0.05 ^Ba^	<0.010	<0.010	0.207
			14 d	0.96 ± 0.03 ^Ab^	1.10 ± 0.04 ^Ab^	1.46 ± 0.07 ^Aa^			
		MUC2	0 d	0.09 ± 0.01 ^B^	0.10 ± 0.01 ^B^	0.11 ± 0.03 ^B^			
			7 d	0.13 ± 0.02 ^AB^	0.17 ± 0.01 ^B^	0.21 ± 0.02 ^AB^	<0.010	0.077	0.784
			14 d	0.27 ± 0.04 ^A^	0.30 ± 0.04 ^A^	0.35 ± 0.06 ^A^			

Note: ^ab^ Different lowercase letters superscripted in the same row indicate significant differences between different treatments (*p* < 0.05). ^ABC^ The superscript values in the same column with different uppercase letters indicate significant differences at different times (*p* < 0.05). The same letter or no letter indicates no significant difference (*p* > 0.05). There is no meaning between uppercase and lowercase letters.

**Table 8 animals-14-00177-t008:** Effects of different treatments of transport stress on the expression levels of relevant genes in the jejunum and colon.

Items			Time		Groups		*p*-Value		
				CG	NG	EG	Time	Treatment	Time × Treatment
Jejunum	mRNA expression	TRAF6	0 d	3.01 ± 0.15 ^A^	2.82 ± 0.16 ^A^	2.70 ± 0.19 ^A^			
			7 d	2.65 ± 0.20 ^AB^	2.50 ± 0.18 ^A^	2.34 ± 0.31 ^A^	<0.010	0.011	0.765
			14 d	2.26 ± 0.15 ^Ba^	1.82 ± 0.13 ^Bb^	1.52 ± 0.13 ^Bb^			
		TLR4	0 d	2.81 ± 0.15 ^Aa^	2.51 ± 0.16 ^Aab^	2.33 ± 0.16 ^Ab^			
			7 d	2.43 ± 0.13 ^ABa^	2.17 ± 0.11 ^ABab^	1.90 ± 0.12 ^ABb^	<0.010	<0.010	0.998
			14 d	2.10 ± 0.19 ^B^	1.86 ± 0.16 ^B^	1.67 ± 0.21 ^B^			
		MyD88	0 d	3.03 ± 0.15 ^Aa^	2.74 ± 0.15 ^Aab^	2.36 ± 0.23 ^Ab^			
			7 d	2.71 ± 0.12 ^ABa^	2.41 ± 0.15 ^ABab^	2.13 ± 0.17 ^ABb^	<0.010	<0.010	0.997
			14 d	2.33 ± 0.17 ^Ba^	2.07 ± 0.13 ^Bab^	1.69 ± 0.16 ^Bb^			
		NF-kB	0 d	2.92 ± 0.16 ^A^	2.50 ± 0.22	2.56 ± 0.12 ^A^			
			7 d	2.54 ± 0.18 ^ABa^	2.15 ± 0.17 ^ab^	1.83 ± 0.13 ^Bb^	<0.010	<0.010	0.556
			14 d	2.06 ± 0.18 ^Ba^	1.97 ± 0.18 ^ab^	1.54 ± 0.16 ^Bb^			
	Protein expression	TLR4	0 d	0.53 ± 0.04 ^Aa^	0.38 ± 0.06 ^b^	0.45 ± 0.01 ^Aab^			
			7 d	0.37 ± 0.04 ^B^	0.29 ± 0.04	0.26 ± 0.06 ^B^	<0.010	0.028	0.774
			14 d	0.27 ± 0.05 ^B^	0.19 ± 0.09	0.14 ± 0.01 ^B^			
		NF-kB p65	0 d	0.94 ± 0.05 ^A^	0.89 ± 0.02 ^A^	0.74 ± 0.08 ^A^			
			7 d	0.72 ± 0.07 ^Ba^	0.64 ± 0.05 ^Bab^	0.50 ± 0.07 ^Bb^	<0.010	<0.010	0.799
			14 d	0.67 ± 0.04 ^Ba^	0.49 ± 0.05 ^Cb^	0.38 ± 0.05 ^Bb^			
		MyD88	0 d	0.21 ± 0.02 ^A^	0.15 ± 0.04 ^A^	0.20 ± 0.01 ^A^			
			7 d	0.17 ± 0.03 ^A^	0.14 ± 0.04 ^AB^	0.13 ± 0.02 ^B^	<0.010	0.052	0.662
			14 d	0.09 ± 0.005 ^B^	0.04 ± 0.01 ^B^	0.06 ± 0.004 ^C^			
		TRAF6	0 d	2.66 ± 0.12 ^A^	2.41 ± 0.17	2.28 ± 0.15 ^A^			
			7 d	2.49 ± 0.14 ^ABa^	2.23 ± 0.13 ^ab^	1.96 ± 0.12 ^ABb^	<0.010	<0.010	0.981
			14 d	2.13 ± 0.17 ^B^	1.95 ± 0.18	1.72 ± 0.13 ^B^			
Colon	mRNA expression	TLR4	0 d	3.07 ± 0.13 ^Aa^	2.60 ± 0.17 ^Ab^	2.37 ± 0.17 ^Ab^			
			7 d	2.65 ± 0.19 ^ABa^	2.29 ± 0.20 ^ABab^	1.95 ± 0.17 ^ABb^	<0.010	<0.010	0.998
			14 d	2.22 ± 0.21 ^Ba^	1.82 ± 0.19 ^Bab^	1.54 ± 0.14 ^Bb^			
		MyD88	0 d	2.86 ± 0.14	2.71 ± 0.10	2.74 ± 0.15			
			7 d	2.76 ± 0.17	2.61 ± 0.12	2.64 ± 0.18	0.432	0.706	0.999
			14 d	2.61 ± 0.14	2.52 ± 0.22	2.59 ± 0.34			
		NF-kB	0 d	2.91 ± 0.17 ^A^	2.71 ± 0.15 ^A^	2.51 ± 0.10 ^A^			
			7 d	2.58 ± 0.12 ^ABa^	2.34 ± 0.18 ^Aab^	1.93 ± 0.13 ^Bb^	<0.010	<0.01	0.903
			14 d	2.18 ± 0.19 ^Ba^	1.82 ± 0.10 ^Bab^	1.55 ± 0.20 ^Bb^			
	Protein expression	TLR4	0 d	0.20 ± 0.01 ^A^	0.18 ± 0.05 ^A^	0.20 ± 0.01 ^A^			
			7 d	0.12 ± 0.02 ^B^	0.10 ± 0.02 ^AB^	0.10 ± 0.01 ^B^	<0.010	0.104	0.505
			14 d	0.09 ± 0.003 ^B^	0.05 ± 0.01 ^B^	0.02 ± 0.003 ^C^			
		NF-kB p65	0 d	1.07 ± 0.10 ^A^	0.98 ± 0.11	0.94 ± 0.10			
			7 d	0.95 ± 0.05 ^AB^	0.84 ± 0.08	0.78 ± 0.09	<0.010	0.184	0.916
			14 d	0.73 ± 0.07 ^B^	0.77 ± 0.04	0.62 ± 0.09			
		MyD88	0 d	0.38 ± 0.05 ^A^	0.30 ± 0.05 ^A^	0.33 ± 0.03 ^A^			
			7 d	0.24 ± 0.04 ^AB^	0.19 ± 0.04 ^AB^	0.22 ± 0.05 ^AB^	<0.010	0.193	0.996
			14 d	0.18 ± 0.05 ^B^	0.11 ± 0.02 ^B^	0.15 ± 0.04 ^B^			

Note: ^ab^ Different lowercase letters superscripted in the same row indicate significant differences between different treatments (*p* < 0.05). ^ABC^ The superscript values in the same column with different uppercase letters indicate significant differences at different times (*p* < 0.05). The same letter or no letter indicates no significant difference (*p* > 0.05). There is no meaning between uppercase and lowercase letters.

## Data Availability

The data that support the findings of this study are available from the corresponding author at any time upon reasonable request. The original sequences we used for the primer sequence design can be found in GenBank (https://www.ncbi.nlm.nih.gov/genbank/, accessed on 18 November 2021.) under the accession numbers XM_042235171, >XM_042233706.1, >XM_027976040.2, >DQ152978.1, NM-102179831, XM_015101111.3, >NM_001135930.1, >DQ152979.1, and >XM_027966801.2.

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
