# Peer review of "Effects of Electrolyte Multivitamins and Neomycin on Immunity and Intestinal Barrier Function in Transported Lambs"

_animals, 2024, doi:10.3390/ani14020177_

Round 1

Reviewer 1 Report (Previous Reviewer 2)

Comments and Suggestions for Authors

Dear Author(s),

Please revise the manuscript as in the attachment. In addition, please re-construct your manuscript according to the journal rules. 

Best Regards.

Author Response

Reviewer 2 Report (Previous Reviewer 1)

Comments and Suggestions for Authors

Animals_ Transportation stress in lambs.

The manuscript titled Effects of different treatments of transport stress on immune function and intestinal health of lambs by Xia and co-workers subjected 60 lambs to eight hours of transportation with treatment of electrolytic vitamins, neomycin, and control prior to transport and the days following. Five lambs per treatment were collected following stress on day 0, then again on day 7 and 14 post transport. Blood and tissues were evaluated.

All animals are not accounted for in the treatments and collection period. It seems 15 animals were collected from each treatment group—5 on each days 0, 7, and 14. But there are 20 animals per group. All measures are done on the animals that were to be collected on those days. There are 15 animals unaccounted for. Perhaps the experimental n = 45 and not 60.

Major concerns are the presentation of results. There are no treatment by time interactions yet all the data are presented as if there were. Data should be presented as treatment difference without reference to time. That there was change across time is expected. There was a stress and they recovered from it. Tables should only have treatment differences, and time differences can be presented in the text. This will make the results more clear for the reader. Also suggest the authors get away from the terminology “significant” and “extremely significant”. They are just different. A small p value does not imply a greater difference—there is just more confidence in that difference. The text and table results are redundant and confusing. Please revise for clarity.

Table 6 could be presented in the text. Treatment had minimal effects on VFA—other than Propionate.

It is speculation to suggest that prolonging treatment would enhance results. Since animals were not treated for 14 days you will not know what would happen if they were.

The discussion is mostly a reiteration of the results. Instead of being redundant please do a better job of discussing the meaning of those results. I am unsure why there was a more consistent result with the electrolytic vitamins than neomyacin. This needs to be discussed.

Since there are no treatment by time interactions Figs 1 & 2 have excessive representatives. Suggest showing only 1 from each treatment. Figure 3 and 4 also no treatment by time interaction. MUC2 in the jejunum does not differ with treatment. A representative sample of the western blots is acceptable. The interpretation +/- is confusing and does not align with the results presented in the table.

Other considerations.

Line 11. “Stress” rather than “It”

Line 17. Specify how stress and treatment affected results—instead of just stating they did.

Line 23. “In sheep production” is not needed

Line 24. Delete “seriously” to read “…after they are transported, affecting their health and welfare.” Delete “of animals”

Line 31. Blood was also collected on each collection day. Delete “And” suggest starting “Liver, spleen, jejunal…”

Line 33. Specify change.

Line 47. Suggest “suppression of the immune system” rather than “immunity decreasing”.

Line 77. Add the source of the electrolytic vitamin and neomycin. I know it is provided later in the text—but it is best to state the source at the first mention of treatment. Delete it from line 89 and 90.

Line 111. Suggest year is not relevant. Suggest deleting. Specify northern hemisphere.

Line 116. Suggest “food and water was not provided” and “animals were allowed to drink within 12 hours following transport.” rather than “were prohibited” and “returning to the enclosure after…”

Line 126. Suggest “…samples were thawed and mixed prior to testing.”

Line 190. Suggest “Effects P0.05 were considered different.”

Line 245. There is only a tendency P = 0.07 for treatment by time interaction for V/C in the jejunum. This must be specified.

Line 335. Redundant with results.

Line 339. It is not clear what is meant by “body state of lambs”

Line 345. “Lead” rather than “Leaded”. Clarify what is meant by “ increase in vitamin consumption”

Line 347. Clarify what is meant by “related to individual differences.”

Line 360 and throughout. It is not clear what is meant by “extension of feeding time”. I think this is just the time effect. The animals recover from the stress with time. It is simply time and not “feeding time”

Line 370. Results are repeated.

Line 373. See above (line 360)

Line 388. Provide citation for a better protective effect

Line 391. Supporting evidence is necessary for “effectively enhance the disease resistance…”

Line 393. I don’t think IgM is “consumed”. Please revise for clarity.

Line 400 – 401, 420, 432, 453. Speculation. Please delete.

Line 412-413. See above (line 360).

Line 436. Provide evidence as to why it would be more effective.

Line 449. Since tight junctions were not assessed in the current study—you can only say it would “potentially enhance tight junctions…”

Line 456. Same as above. Not directly measured. Potential difference is all that can be inferred.

Line 457. See above (line 360).

I don’t believe reference #21 is cited appropriately. It has to do with ascorbic acid and chickens.

Comments on the Quality of English Language

The manuscript must be edited. It is difficult to read--mostly because it is redundant and presented not according to the significance.

Round 2

Reviewer 1 Report (Previous Reviewer 2)

Comments and Suggestions for Authors

Dear Author(s),

I see that you meet almost all the revisions mentioned in the study. However, in the "response to reviewers" file, you stated as the 11th item "Line 191 Please provide standard error of these features." I couldn't see an answer about it. The standard error is the value that should be given next to the mean as a result of ANOVA. I expect you to revise this.

Sincerely yours

Author Response

Dear reviewer,

  Thank you very much for your comments and professional advice. These opinions help us to improve our academic rigor of article. Based on your suggestions, we have tried our best to polish the language and adjust the manuscript “Effects of different treatments on immune function and intestinal health of transported lambs. (ID: animals-2764160)”. We hope that our work can be improved again. The details as follows:

Reviewer#:

I see that you meet almost all the revisions mentioned in the study. However, in the "response to reviewers" file, you stated as the 11th item "Line 191 Please provide standard error of these features." I couldn't see an answer about it. The standard error is the value that should be given next to the mean as a result of ANOVA. I expect you to revise this.

Answer: Thanks for your comment. We apologize for our carelessness. The standard error was provided in the manuscript and highlight it in red. You can view these contents in the manuscript.

Thank you very much for your attention and time. Look forward to hearing from you.

Yours sincerely,

Cui Xia

23 December 2023.

This manuscript is a resubmission of an earlier submission. The following is a list of the peer review reports and author responses from that submission.

Round 1

Reviewer 1 Report

Comments and Suggestions for Authors

The manuscript titled Effects of different treatments of transport stress on immune function and intestinal health of lambs requires editing before it can even be appropriately reviewed. The manuscript lacking line and page numbers prevents the reviewer from providing line by line feedback.

The biggest concerns are in the results the text is redundant with the tables. This makes the manuscript long and difficult to read. The discussion is too long and a reiteration of the results. Results are over-interpreted (eg. a small change in proprionate in the colon is now "differences in VFA"

I am confused as to why it is an electrolytic vitamin since the minerals provided are commonly considered "electrolytes".

Trial and trail and different words. Make sure you are using them appropriately.

I think your standard is beta-actin not beta-action.

In the abstract report the direction of change and not only that things differed.

Significance is a measure of confidence that the differences reported are real. A smaller p value gives you more confidence that the difference is real--it isn't more significantly different.

Table 1 organize your composition in order of greatest to least content.

It is confusing in the methods if the lambs were co-housed as stated in the first paragraph or in different pens as stated om section 2.2

Are liver and spleen weights per body weight really an index? It seems they are organ weight per body weight.

COR --is not defined.

Slaughtered is the wrong word. It suggests either they were harvested for food (butchered), or they were wrongly killed. Suggest- lambs were anesthetized, exsanguinated, and tissues were collected.

Reference 14 is a fish paper. I don't think it should be used in support of mammals--especially without specification.

You cannot conclude that your vitamin treatment is "most effective"--only amongst treatments you presently investigated. There are infinite treatments that may prove to be better. This needs to be tempered.

The western blots need to show treatments, not only the interpretation.

Comments on the Quality of English Language

This needs serious editing. Good luck.

Reviewer 2 Report

Comments and Suggestions for Authors

Dear Author(s),

Please make the revision of the manuscript.

Best Regards.
